# Functional orderly topography of brain networks associated with gene expression heterogeneity

Wei Liu[1], Ling-Li Zeng [1], Hui Shen[1], Zong-Tan Zhou[1] & Dewen Hu [1✉]

The human cerebral cortex is vastly expanded relative to nonhuman primates and rodents, leading to a functional orderly topography of brain networks. Here, we show that functional topography may be associated with gene expression heterogeneity. The neocortex exhibits greater heterogeneity in gene expression, with a lower expression of housekeeping genes, a longer mean path length, fewer clusters, and a lower degree of ordering in networks than archicortical and subcortical areas in human, rhesus macaque, and mouse brains. In particular, the cerebellar cortex displays greater heterogeneity in gene expression than cerebellar deep nuclei in the human brain, but not in the mouse brain, corresponding to the emergence of novel functions in the human cerebellar cortex. Moreover, the cortical areas with greater heterogeneity, primarily located in the multimodal association cortex, tend to express genes with higher evolutionary rates and exhibit a higher degree of functional connectivity measured by resting-state fMRI, implying that such a spatial distribution of gene expression may be shaped by evolution and is favourable for the specialization of higher cognitive functions. Together, the cross-species imaging and genetic findings may provide convergent evidence to support the association between the orderly topography of brain function networks and gene expression.

[1] College of Intelligence Science and Technology, National University of Defense Technology, Changsha, Hunan 410073, P. R. China. ✉email: dwhu@nudt.edu.cn

As the fruit of billions of years of evolution, the human brain has developed a hierarchical organizational infrastructure that remains similar to some extent among mammals, including the neocortex, archicortex, and subcortical areas[1,2]. On this basis, almost all mammalian brains demonstrate orderly topography in their functional networks, in which higher cognitive functions are primarily regulated by the neocortex rather than the archicortical and subcortical areas[3,4]. The spatial and topological layout of functional brain networks are highly heritable, likely due to genome blueprint[5–7]. However, the underlying molecular architecture that supports such functional topography of brain structures is poorly understood.

While the basic layout of cortical areas can be traced back to the homologous brain regions in rodents and non-human primates, the human brain has undergone remarkable changes in the process of evolution, with the human cerebral cortex being vastly expanded relative to non-human primates and rodents and disproportionately composed of multimodal association areas[8,9]. The multimodal association areas and their structural and functional connections in the human brain, primarily constituting cognitive functional networks, such as the frontoparietal network, salience network, and default-mode network, play an essential role in higher-order brain functions[10–12]. Most recent studies suggest that the development of higher-order cognitive networks in recent human brain evolution is associated with specific gene expression profiles[13–15]. The highly consistent transcriptional architecture in neocortex is correlated with resting-state functional connectivity[13], and genetic and evolutionary uncoupling of structure and function in different transmodal systems may support the emergence of complex forms of cognition[15]. However, the pattern of gene expression in the human brain to support the emergence of higher-order brain functions remains to be determined.

The availability of genome-wide spatial patterns of gene expression data provide a great opportunity to understand the relationship between gene expression and the anatomical and functional organization of the mammalian brain[16–19]. Over the past decade, a number of transcriptome studies focusing on human brain region-related changes in gene expression profiles have been published. Recent studies performing resting-state functional magnetic resonance imaging (fMRI) have revealed that functional connectivity networks can be recapitulated using correlated gene expression in post-mortem brain samples[20]. Further studies reported that some genes could influence connectivity strength between network hubs of the human connectome[21], and human-accelerated genes could play a role in the expansion of higher-order cognitive networks[7] using comparative transcriptomics analysis. These findings imply that the emergence of orderly functional networks in the human brain may be the result of changes at the genetic level.

We hypothesized that there might be gene expression heterogeneity in various brain structures associated with the functional orderly topography of brain networks. Here, we measured gene expression heterogeneity with two metrics, i.e., the proportion of housekeeping genes occupying all expressed genes and topology indices of the gene expression networks. Housekeeping genes (HKGs) are usually expressed at relatively constant rates for the basic molecular and cellular function of neurons[22,23]; thus, the lower proportion of HKGs expressed in brain structures may suggest their higher functional diversity. Topological indices of the gene expression networks in brain structures, including mean path length, clustering coefficient and eigen entropy, offer possibilities to understand how genes work together to perform diverse functions[24]. In particular, we compared the heterogeneity of gene expression in multimodal cortex with that of unimodal cortex to investigate the role of gene expression in the specialization of higher-order cognitive function in human brain evolution. Then, we studied the association of the expression heterogeneity of brain structures and the hierarchical architecture of functional connectivity networks to reveal the basic principle by which the brain is organized throughout evolutionary history, at least across rodents, rhesus macaques, and humans.

## Results

**The proportion of HKG expression is lower in the neocortex than in the archicortex and subcortex**. We constructed the gene expression networks for all brain samples by combining human gene expression data from the Allen Institute for Brain Science (http://human.brain-map.org/) and large-scale protein interaction data (see the Methods section). The gene expression data were obtained from six adult brains (two included both hemispheres, and four included one hemisphere) for a total of 3702 brain samples[25]. Considering individual differences and different numbers of samples among donors, we analysed and compared the gene expression networks between human brain samples. Based on the gene expression networks, we extracted the percentages of HKGs and specific genes that were expressed in almost all brain samples[26] and expressed in only one or two samples[27], respectively. Although the values of these indicators are variable in different donors, their distribution in the neocortex to archicortex and subcortex are basically the same across all donors. We took Brain #1 as an example to show the analysis results, since it has the highest number of samples of all donors.

The percentages of HKGs are mapped to the samples of Brain #1 (Fig. 1a). The samples with relatively low percentages of HKGs were primarily located in the neocortex of the human brain. Similar results were obtained in the other five human brain samples (Supplementary Fig. 1 and Supplementary Data 1). As shown in Fig. 1b, in Brain #1, the HKG percentages in the neocortex were significantly lower than those in the archicortex and subcortex (both P-values < 0.001 in two sample t-tests). Not surprisingly, the genes expressed in the neocortex tended to have lower gene expression levels and higher expression specificity than those in the archicortex and subcortex (Supplementary Data 1).

The HKG percentages were mapped to the main structures of Brain #1 (Fig. 1c). Interestingly, the cerebellum exhibited a relatively lower mean HKG percentage than the cerebral nuclei, interbrain, and brainstem. However, after dividing the cerebellum by interior structures, significant differences were observed in the HKG percentages between the cerebellar cortex and cerebellar deep nuclei. The cerebellar cortex, including the lateral hemisphere ($33.60 \pm 1.58\%$) and paravermis ($33.61 \pm 1.76\%$), exhibited a lower HKG percentage than the cerebellar deep nuclei ($39.70 \pm 4.74\%$). Thus, significant differences in expression may exist between the different subregions of the cerebellum, coinciding with their functions. According to previous studies, the cerebellum in humans not only plays an important role in motor control, which is mainly executed by the cerebellar deep nuclei, but is also involved in certain higher-order cognitive functions, such as attention and language, which are executed by the cerebellar cortex[28,29].

To investigate the relationship of the HKG percentage and evolution, we analysed the evolutionary rate of gene expression (see the Methods section) to measure the selective constraints on the brain regions involved. As shown in Fig. 1d, the HKG percentages in the samples of Brain #1 were found to decrease as the average evolutionary rates of gene expression increased. Similar results were observed in the samples of the other five brains. These results imply that the brain regions with lower HKG

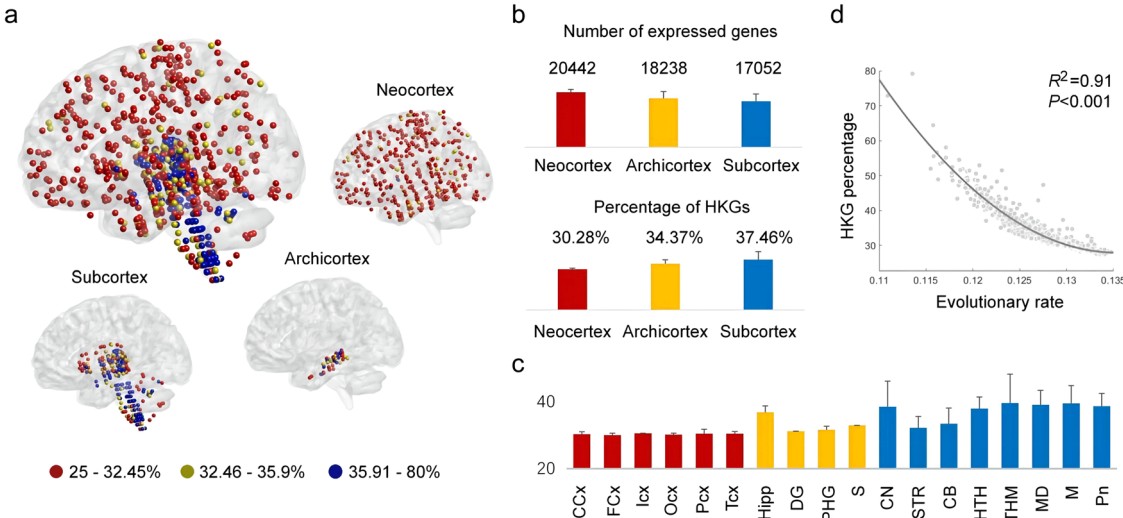

**Fig. 1 Gene expression characteristics of human brain samples. a** The percentage of HKGs mapped to the brain samples of Brain #1. The proportion of HKG expression to that of other specific genes was considerably higher in the archicortex and subcortex than in the neocortical areas. The HKG percentage cut-off to best distinguish the neocortex and the archicortex and subcortex of Brain #1 was set at 32.45% and 35.90%, respectively. **b** The mean value and standard deviation of the number of expressed genes and HKG percentage in the neocortex, archicortex and subcortex of Brain #1. **c** The percentage of HKGs mapped to the main structures of Brain #1. The structures in the neocortex, archicortex and subcortex are marked in red, yellow and blue, respectively. CCx cingulate neocortex, FCx frontal neocortex, Icx insular neocortex, Ocx occipital neocortex, PCx parietal neocortex, TCx temporal neocortex, Hipp hippocampal proper, DG dentate area, PHG parahippocampal gyrus, S subiculum, CN cerebral nuclei, STR striatum, CB cerebellum, HTH hypothalamus, THM thalamus, MD medulla, M midbrain, Pn pons. **d** The average evolutionary rate of gene expression was negatively correlated with the HKG percentage in the samples of Brain #1.

percentages show a higher average evolutionary rate under less evolutionary pressure than the other regions.

**Newly developed brain regions exhibit increased heterogeneity in their internal expression networks**. Considering that genes usually function together through the interaction of their products, we hypothesized that the heterogeneity of gene expression in different brain structures may affect the organizational structure of the gene expression networks, thus showing different topological properties. To analyse the gene expression heterogeneity from a network perspective, we computed three typical topological indices of the gene expression networks, including the mean path length, clustering coefficient and eigen entropy, to measure the overall navigability, modularity, and order of the networks (see the Methods section and Supplementary Data 2). In general, a longer mean path length, smaller clustering coefficient and larger eigen entropy of a network may suggest that it is sparser, contains fewer clusters and is more disorderly, and thus shows greater heterogeneity of gene expression.

The gene expression networks in the neocortex exhibited greater heterogeneity, with longer mean path lengths, smaller clustering coefficients and larger eigen entropies, than those in the archicortex and subcortex (Fig. 2a, b). Meanwhile, significant correlations were observed among the three topological indices and the average evolutionary rate of genes. We found that the mean path lengths in the brain samples increased as the average evolutionary rates of genes increased, as shown in Fig. 2c. Our results revealed that the orderness of the gene expression networks in the brain regions presented a downwards trend from the subcortex to the archicortex to the neocortex, consistent in all six brains. Previous studies have reported that the eigen entropy of the whole-brain network in the human connectome is associated with the neurodevelopment and ageing of individuals[30]. Considering the theory of evolution of life systems that the development of organisms and a highly complex brain is a process of diminishing entropy[30,31], such brain organization may be the result of evolution.

**Brain regions with greater expression heterogeneity were primarily located in the multimodal association cortex of the human brain**. To investigate the heterogeneity of interconnected networks in the neocortex, we integrated the gene expression data of brain samples from six adult individuals through the standard process described in previous studies[32] (see Supplementary Methods for details) and mapped them to seven networks of the cerebral cortex[33]. According to the gene expression heterogeneity, seven brain networks showed orderly topography from the multimodal association cortex to the unimodal primary cortex. The multimodal association cortex, including frontoparietal control, attention, and default networks, showed reduced gene expression similarity within networks compared to the visual, motor, and limbic networks (Fig. 3a, see the Methods section), while the higher-order cognitive functional networks consistently exhibited larger differences in gene expression than the visual and motor networks (Fig. 3b). Meanwhile, the gene expression heterogeneity within networks was found to be positively correlated with the corresponding standard error of gene expression levels ($R = 0.87$, $P = 0.01$). The multimodal association cortex contained a lower standard error of gene expression levels than the unimodal primary cortex, implying that its gene expression is more heterogenous, as shown in Fig. 3c. These results indicated that the multimodal association cortex exhibited greater expression heterogeneity, not only in the lower expression similarity within networks but also in the greater dispersion degree of gene expression levels.

**Brain regions with greater expression heterogeneity tended to be connected to more regions in the functional connectivity networks**. To explore whether genes with highly consistent cortical patterns across individuals drive this functional organization, we compared the gene expression with resting-state functional connectivity MRI data from the Human Connectome Project[34,35]. We generated a region-level functional connectivity matrix $C$ averaged across 50 subjects using linear correlations of 116 regions from the

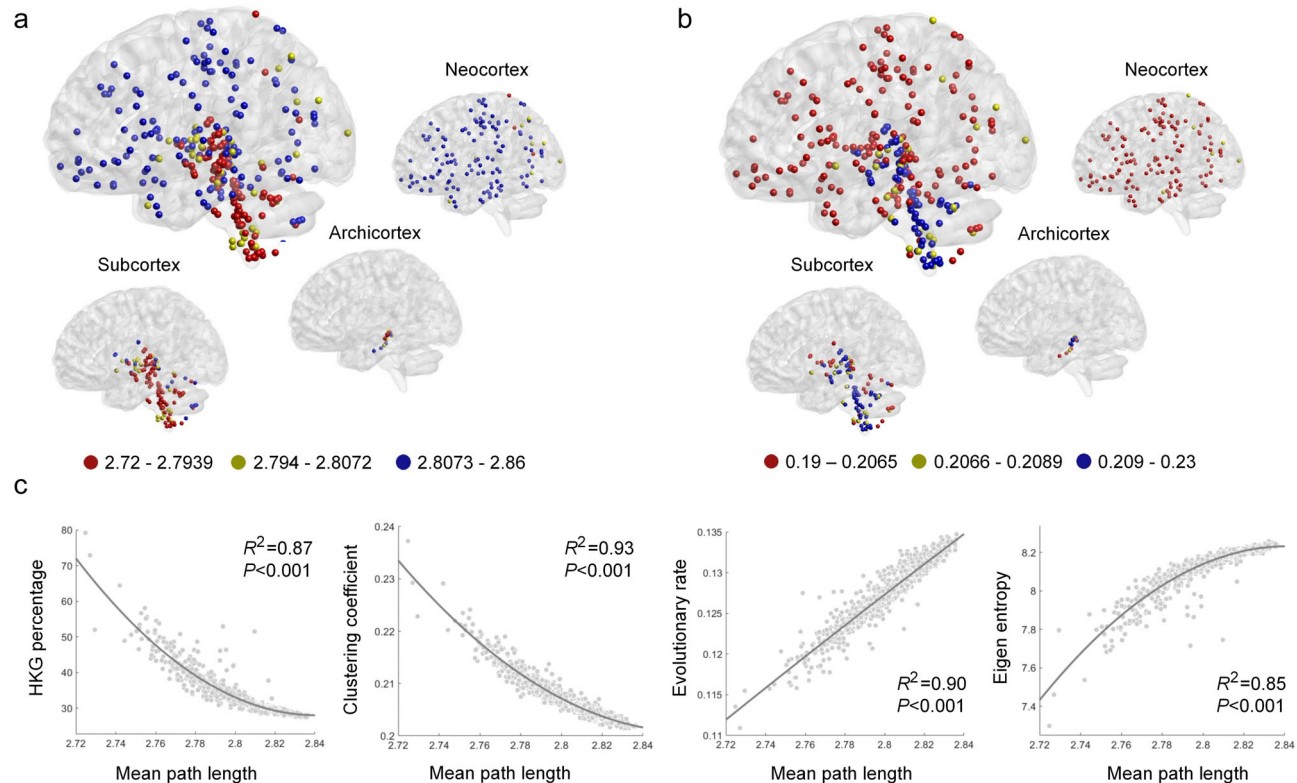

**Fig. 2 Topological properties and the relationship of the gene expression networks. a** The mean path length and **b** clustering coefficient mapped to all brain structures in Brain #1. The gene expression networks in the neocortex have longer mean path lengths and smaller clustering coefficients than those in the archicortex and subcortex. The mean path length cut-offs to best distinguish the neocortex, archicortex and subcortex of Brain #1 are 2.8073 and 2.7939, respectively, while the cut-offs of the clustering coefficient are 0.2065 and 0.2089, respectively. **c** The mean path length was negatively correlated with the HKG percentage and clustering coefficient and positively correlated with the evolutionary rate of expressed genes and eigen entropy of the gene expression networks in the samples of Brain #1.

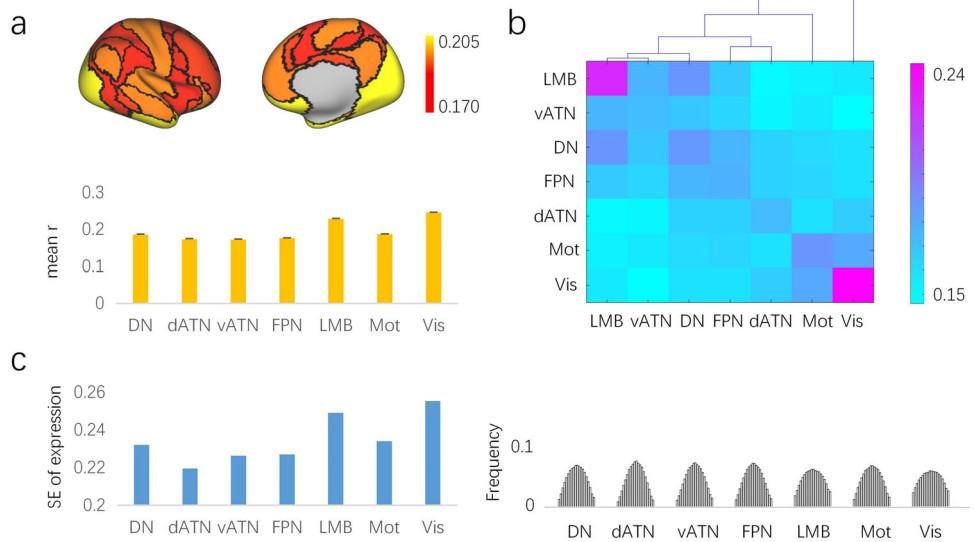

**Fig. 3 The expression heterogeneity of seven networks in the human cerebral cortex. a** Mean correlation coefficient of gene expression of samples within each network. **b** The hierarchical clustering of gene expression between networks. **c** The standard error of gene expression levels in samples of each network and their corresponding distribution. DN default-mode network, dATN dorsal attention network, vATN ventral attention network, FPN frontal parietal network, LMB limbic network, Mot motion network, Vis visual network.

automated anatomical labelling (AAL) atlas[36] (see the Methods section). Then, we computed the topological properties of the functional connectivity networks in the human brain, including the degree and betweenness coefficient of each brain region.

We compared the properties of the gene expression networks with those of the functional connectivity networks in the human brain. The degree of brain regions in the functional connectivity networks was negatively correlated with the HKG percentage

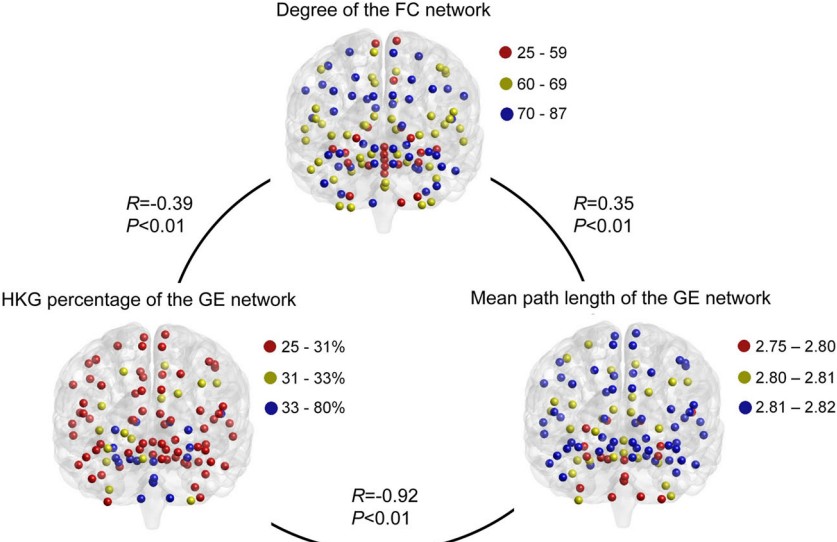

**Fig. 4 Comparison of the gene expression networks and the functional connectivity network in human brains.** The degree of the functional connectivity network was significantly negatively correlated with the HKG percentage and positively correlated with the mean path length of the gene expression network in Brain #1. FC functional connectivity, GE gene expression.

($R = -0.39$, $-0.25$, $-0.31$, $-0.20$, $-0.20$, and $-0.33$ for Brains #1–6, respectively, all with $P < 0.01$) and positively correlated with the mean path length of the gene expression network ($R = 0.35$, 0.31, 0.37, 0.16, 0.25, and 0.32 for Brains #1–6, respectively, $P < 0.01$). Thus, brain regions densely connected to other regions in the functional connectivity networks are primarily located in the neocortex of the human brain and tend to exhibit greater expression heterogeneity, with lower proportions of HKGs and longer mean path lengths in their expression networks (Fig. 4 and Supplementary Fig. 2). These results suggested that the brain regions with greater expression heterogeneity tend to be closely related to each other to perform more interdependent functions.

**The gene expression patterns in the mouse and rhesus macaque brain are similar to the human brain.** Subsequently, based on the expression data from the Allen Institute mouse brain atlas from a 56-day-old male C57BL/6 J mouse brain[37] and protein interactions in the mouse brain, we established the gene expression networks in 73 structures of the adult mouse brain (see the Method and Supplementary Data 3). We identified 570 HKGs that were expressed in 72 or 73 structures. Most of the HKGs (33.68–79.65%) in the mouse brain are homologous to the HKGs in human Brains #1–6. Mouse brain regions with lower HKG proportions tended to be newly developed, as shown in Fig. 5a. Focusing on the isocortex of the mouse brain, the 'perirhinal area' exhibited the lowest percentage of HKGs, followed by the 'prelimbic area', with values of 29.17% and 30.76% (Supplementary Fig. 3), respectively, consistent with those of previous evolutionary analyses across species[38]. In particular, the mouse cerebellum has the highest HKG proportion (44.90%), suggesting that there may be significant evolutionary differences in the cerebellum between humans and mice[39]. We found that the mean path length and eigen entropy of the gene expression networks in the isocortex ($4.09 \pm 0.09$ and $5.13 \pm 0.10$, respectively) were significantly higher than those in the cerebral nuclei ($3.83 \pm 0.10$ and $4.96 \pm 0.05$, respectively), indicating that the brain structures that evolved later exhibit greater heterogeneity than those that evolved earlier. The cerebral nuclei, including striatum and pallidum, is a part of mouse cerebrum. Based on resting-state functional MRI data of forty-eight male C57BL/6 J mice[40,41], we established the functional connectivity networks of the mouse brains (see the

Methods section) and found a negative correlation between the HKG percentage in gene expression and region degree in functional connectivity networks ($R = -0.53$, $P = 0.01$, Fig. 5b), confirming the results obtained from the human brains.

Furthermore, we generated the expression networks by integrating the gene expression data of the adult rhesus macaque specimen from the NIH Blueprint Non-Human Primate Atlas[42] with interaction data from the STRING database and the functional connectivity networks based on monkey functional MRI data (see Method and Supplementary Data 4). We found that the gene expression networks in the neocortex tended to have greater heterogeneity, with a lower HKG percentage, longer mean path length and higher functional degree, than those in the archicortex and subcortex (Fig. 5c, d), in line with the results obtained from the human brain.

**Discussion**

Our combined comparative neuroimaging and genetic findings suggest that heterogeneous changes in gene expression may play an important role in the formation of the functional topography in the human brain. Based on the multiresolution gene expression networks in human brain samples, and the structure and region levels in rhesus macaque and mouse brains, the heterogeneity of gene expression in the neocortex was found to be significantly greater than those in the archicortex and subcortex, associated with the functional orderly topography of brain networks. The results obtained from the different resolution networks and different individuals are consistent, implying that such gene expression patterns in the brain are robust and inherent. Based on the abagen toolbox[43], we examined the influence of methodological variability on our results and found that the change trend of the expression heterogeneity from the neocortex to the subcortex is consistent under different standardizing workflows (see Supplementary Methods and Supplementary Data 5). In particular, the increased heterogeneity of gene expression in multimodal association areas may potentially regulate the specialization of higher-order cognitive functions in human brain evolution, compatible with prior observations of high expression of evolution-related genes in these brain areas[7] and increased transcriptional complexity in the frontal lobe of the human brain[44].

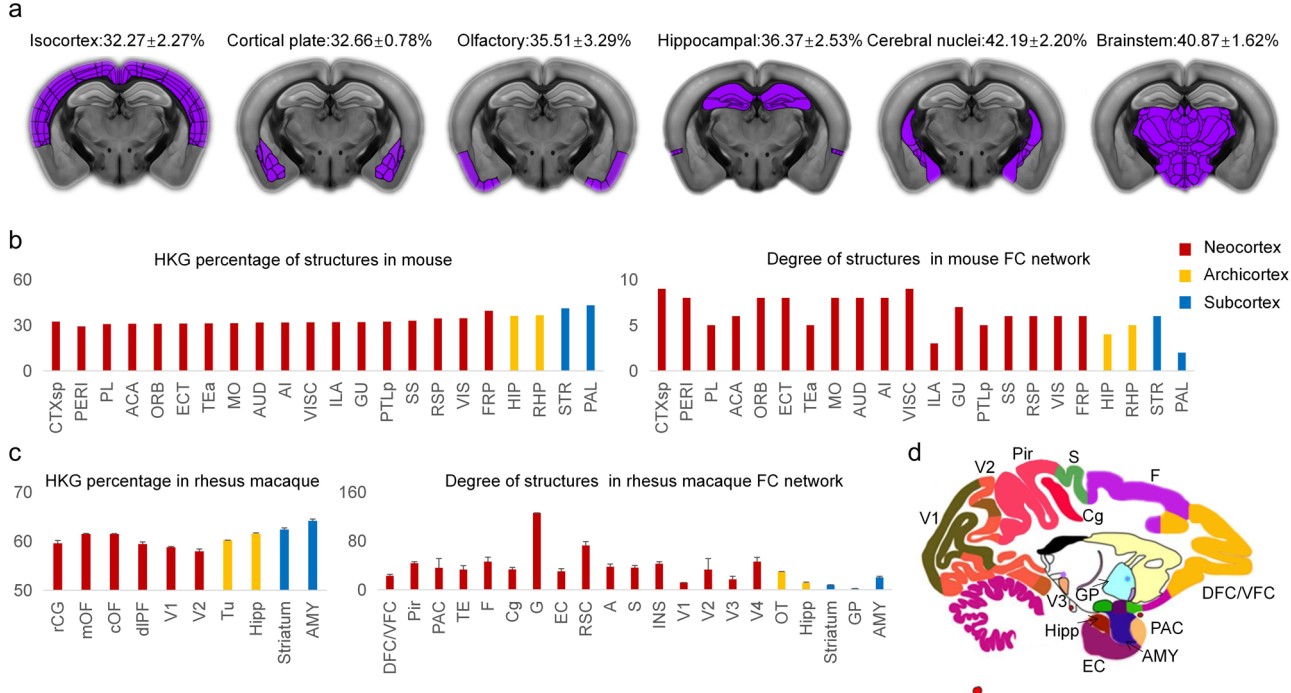

**Fig. 5 Comparison of the gene expression network and the functional connectivity network in mouse and rhesus macaque brains. a** The HKG percentages mapped to the regions of the mouse brain. The HKG percentage in the neocortex is lower than those in the archicortex (including the olfactory area and hippocampal formation) and subcortex (cerebral nuclei and brainstem). The regions of interest are marked in purple. All hybridization images were obtained from the Allen Mouse Brain Atlas. **b** HKG percentage of gene expression and the degree of functional connectivity in the structures of the mouse brain. **c** The HKG percentage of gene expression and the degree of functional connectivity in the structures of rhesus macaque brains. **d** Diagram of the main structures in the rhesus macaque brain. This picture is generated based on the D99 brain template[74]. The description of brain structures can be found in Supplementary Methods.

The neocortex is the seat of higher cognitive functions[45] and involved in complex cognitive functions through cortical circuits[46], cortical-subcortical[47] and cortical-cerebellar circuits[48]. Previous studies reported that spatial patterns of gene expression may reflect the hierarchical organization[49] and spatial gradients of intrinsic dynamics in neocortex[50]. Our results indicated that the genes expressed in the neocortex tend to have higher variability, lower gene expression levels and higher expression specificity than those in the archicortex and subcortex. This is because there is a higher proportion of non-HKGs, including specific genes, expressed in the neocortex than in the archicortex and subcortex. As reported, HKGs are involved in basic cell maintenance and, therefore, are expected to maintain constant expression levels in all cells and conditions[51] and have a relatively low evolutionary rate[52]. In contrast, specific genes have more variable expression patterns and reduced expression levels[53]. We found a similar trend of gene expression in the cerebral cortex of the human brain, with higher expression variability of genes in the multimodal association cortex than in the unimodal primary cortex.

Evolutionary changes in structures of the human brain relative to other mammalian brains can arise from the emergence of new genes but more from quantitative expression changes in mRNA[54]. Comparative transcriptome studies of the human and chimpanzee brain indicated that the acceleration signal is clearly more pronounced in the PFC, a region involved in high order, partly human-specific cognitive processes such as abstract thinking and planning, than in other brain regions[55]. In our studies, the increased heterogeneity of the gene expression networks from the archicortex and subcortex to neocortex was found to be closely correlated with the average evolutionary rate of expressed genes, implying that the spatial architecture of gene expression is very likely the product of natural selection. Similarly, its associated functional orderly topography of the brain is commonly believed to be shaped under the pressure of evolution[11,56]. The structures in the neocortex expressed more non-HKGs to perform their complex functions under short-term evolutionary pressure, while the structures in the archicortex and subcortex reduced their proportion of specifically expressed genes to maintain the stability of their functions under long-term evolutionary pressure. Such an organization mode of brain networks may contribute to the functions of neocortical areas being implemented effectively and flexibly, while the critical functions of archicortical and subcortical areas are performed steadily and systematically.

Characterization of the human brain from a network perspective has become a powerful tool for inspecting the structural and functional architectures of the brain[57,58]. Previous findings have suggested the large-scale network organization of the human connectome can enable the efficient processing of information and thus support complex brain functions[10,21,59–61]. Weighted gene co-expression network analysis was applied to build co-expression networks, so as to identify modules of co-regulated genes[16] or examine the systems level organization of lineage-specific gene expression differences[44]. Cellular network has topological robustness against accidental failures[62] and the analysis based on gene connectivity may observe the overall conservation of gene co-expression modules between the species[63]. Our results further provide insights into the molecular bases of brain organization and put the changes in gene expression heterogeneity observed into a systems level context. Our analyses showed that increased expression of non-HKGs in the neocortex was associated with changes in the topological properties of the gene expression networks. The gene expression networks in the

neocortex are sparser, contain fewer clusters and are more disorderly than those in the archicortex and subcortex. The distribution of the topological indices of the gene expression networks in the neocortex to archicortex and subcortex was basically consistent across all donors (Supplementary Fig. 4). Such organization possibly contributes to the genes expressed in neocortex involving more diverse functions than those in the archicortex and subcortex at the expense of partial efficiency. The regions in the neocortex exhibited significantly different topological properties both in the gene expression networks and functional connectivity networks compared with those in the archicortex and subcortex to support greater expression heterogeneity in the neocortical areas from the network prospective. The brain regions with greater expression heterogeneity densely connected to other regions in the functional connectivity networks, supporting that the neocortex possesses a mosaic of regions, central to its information-processing capabilities[11]. Meanwhile, our comparative analyses showed that the higher-order cognitive functional networks exhibited greater expression heterogeneity than the sensorimotor networks, in accord with the notion about the rapid expansion of multimodal association areas in the human brain relative to other mammals[56,64].

In this study, we observed a similar orderly topology of gene expression heterogeneity in the human, rhesus macaque, and mouse brains. However, the distributions of gene expression heterogeneity in the brain structures of rhesus macaques and mice are not completely consistent with those of humans. For example, the human cerebellar cortex displayed greater gene expression heterogeneity than the cerebellar deep nuclei, which has not been observed in the mouse brain. Such human-distinct patterns in spatial gene expression coincide with emerging cognitive functions in the human cerebellar cortex[28,29], possibly related to the differences in cognitive abilities between humans and other mammals. Due to the limitation of the sampling fineness of the gene expression dataset in the macaque brain, we have not yet found unique expression characteristics in the macaque brain compared with those in the human brain and mouse brain.

Due to the limitation of brain sampling resolution in different species, this study mainly focused on the global intraspecies differences in gene expression heterogeneity in brain structures, while interspecies regional-matched differences need to be analysed when more datasets are available. At the same time, individual differences were observed in the gene expression characteristics and topological properties between the structures of different individuals within the species. Therefore, differences among brains must be considered in further investigation when the gene expression data of more individuals become available.

## Methods

**Gene expression data of human brains**. Publicly available gene expression data from six human postmortem donors (one female and five males), aged 24–57 years ($M = 42.5$, SD = 13.38), were obtained from the AHBA and downloaded after the updated microarray normalization pipeline implemented in March 2013 (http://human.brain-map.org). The dataset consists of normalized expression data detected by 58,692 probes taken from 3702 spatially distinct tissue samples. Before tissue dissection, donor brains underwent anatomical MRI scanning and alignment to MNI space by the Allen Institute. Available samples were prepared for microarray expression analyses by the Allen Institute via macrodissection for cortical areas or laser dissection for subcortical regions[24]. For additional information on structural imaging data as well as microarray preprocessing and normalization procedures, refer to the AHBA technical white paper (help.brain-map.org/display/humanbrain/Documentation). The expression level of each gene from all probes was averaged if there were multiple probes for the same gene. The resulting dataset contained 29,180 unique mRNA probes, providing transcriptional data of human brains.

**Gene expression networks of human brains**. If two genes were expressed in a given brain sample and available in the integrated human protein interaction network, then both genes were included in the gene expression network of the

sample. The integrated protein interaction dataset was established by merging the previous material[65] with the iRefIndex database[66] (Supplementary Data 6). By integrating the expression data and large-scale protein interaction data, we established the gene expression networks for samples from six human brains. For each sample, we obtained a matrix corresponding to its gene expression network, where nodes represent the genes expressed in the sample and edges represent that their gene products can interact.

**The evolutionary rate of genes**. The evolutionary rate is a measurement used to quantify the speed of evolutionary change. The selective pressure is assumed to be defined by the ratio dN/dS. dS represents the synonymous substitution rate (changing the amino acid), and dN represents the nonsynonymous substitution rate (keeping the amino acid). Under purifying selection, natural selection prevents the replacement of amino acids, so dN will be lower than dS. Values of dN/dS <1, =1, and >1 indicate negative purifying selection, neutral evolution, and positive selection, respectively. We calculated the dN/dS values for all genes expressed in the human brains to characterize their evolutionary rates. The synonymous and nonsynonymous substitution rates between humans and mice were obtained from Ensembl (http://www.ensembl.org/biomart/martview/).

**Topological properties of networks**. Distance in networks is measured with the path length, and the shortest path, the path with the smallest number of links between the selected nodes, has a special role. The mean path length represents the average of the shortest paths between all pairs of nodes and offers a measure of a network's overall navigability[62]. Cellular functions are likely to be carried out in a highly modular manner. The average clustering coefficient characterizes the overall tendency of the nodes to form clusters or groups[67]. The closer the local clustering coefficient is to 1, the more likely that the network will form clusters. The eigen entropy of networks was defined as the entropy of the normalized largest eigenvector of an adjacency matrix[30]. For a network with $n$ genes, its adjacency matrix can be represented as $A_{ij}(i, j = 1 \sim n)$, which describes the interacting relationships of the genes in the networks. If gene $i$ interacts with gene $j$, $A_{ij} = 1$; otherwise, $A_{ij} = 0$. We assumed that the eigenvector of matrix A corresponding to the largest eigenvalue $\lambda_k$ is $r_k$. By dividing by the sum of all eigen vectors, we normalized this vector to obtain its energy concentration, i.e., $I_k = \frac{r_k}{\sum_{i=1}^{n} r_i}$. The eigen entropy of this network is defined as $S_p = -\sum_{k=1}^{n} I_k \ln I_k$, which can reflect the degree of ordering in each network. A smaller eigen entropy of a network corresponds to a higher degree of ordering of the network.

**Gene expression similarity within and between networks**. The similarity of gene expression within a network is defined as the mean value of the correlation coefficient of gene expression between any two samples from this network. The similarity of gene expression between networks is defined as the mean value of the correlation coefficient of gene expression between a pair of samples from two different networks.

**Functional connectivity of the human brains**. Functional connectivity was analysed using resting-state functional MRI data from 50 randomly selected subjects from the Human Connectome Project (HCP, http://www.humanconnectome.org/documentation/S500/). The HCP minimal preprocessing pipeline was used for the resting-state fMRI data[68], which includes artefact removal, motion correction[69] and registration to a standard space (see Supplementary Methods for details). The AAL atlas, which divided the whole brain into 116 regions, was used for the region-to-region functional connectivity measures in the current study[36]. We evaluated the functional connectivity between each pair of regional averaged time courses using Pearson's correlation coefficient and then standardized the functional connectivity matrix with Fisher's Z-transform. Significant functional correlations were selected using one-sample $t$-test ($P < 0.05$, Bonferroni correction), resulting in the binary $116 \times 116$ symmetric connectivity matrix C of the functional connectivity network in human brains (Supplementary Data 7).

**The gene expression network of the mouse brain**. We established the expression networks in mouse brain structures based on the Allen Institute mouse brain atlas, which offers finely sampled whole-genome expression data[37]. According to the Allen Reference Atlas, a 56-day-old male C57BL/6 J mouse brain was partitioned into 73 structures and 12 regions. We computed the expression levels of 719,905 genes in 73 brain structures contained in the coronal planes. Based on the expression intensity in each voxel, we obtained the expression levels of genes in 73 structures by averaging across all voxels in the brain structures. The expression levels in the mouse in situ hybridization data from the Allen Mouse Brain Atlas were quantified using a metric called expression energy (fraction of stained volume × the average intensity of staining) as previously described[37]. In total, 2873 genes were found to be expressed in at least one structure by selecting genes with fractions expressing pixels above 0.02 to omit genes with extremely low expression. We downloaded 33,145 protein interactions among 8499 mouse gene products from the BIOGRID database. By integrating the gene expression data and protein

interactions in the mouse brain, we established interaction networks in 73 structures of the adult mouse brain.

**Functional connectivity of mouse brains**. Resting-state fMRI data of anaesthetized mice were collected from fifty male C57BL/6 J mice (Janvier, Le Genest-St Isle, France) between 10 and 13 weeks old weighing $30.6 \pm 1.9$ g (mean ± SD), which are publicly available on the central.xnat.org repository in Analyze 7.5 format (Project ID: fMRI_ane_mouse). Then, the following steps were performed: (1) slice timing correction; (2) motion correction; (3) normalization with an in-house EPI template; (4) spatial smoothing using a 0.4-mm full width half-maximum Gaussian kernel; (5) linear detrending and bandpass temporal filtering (0.01–0.3 Hz); (6) regression of nuisance variables, including the six parameters obtained by rigid body head motion correction, global signals, and their first temporal derivatives. The functional connectivity between each pair of regional averaged time courses was evaluated using Pearson's correlation coefficient and then standardized to Z scores. Significant functional correlations were selected to obtain the binary $22 \times 22$ symmetric connectivity matrix of the functional connectivity network in mouse brains (Supplementary Data 8).

**Gene expression network of adult rhesus macaque brains**. We downloaded the original gene expression CEL files of the three 48-month specimens generated serially across a complete hemisphere from adult rhesus macaques from the NIH Blueprint Non-Human Primate (NHP) Atlas (http://www.blueprintnhpatlas.org) and extracted normalized and processed the expression levels of all genes based on the R language program. Interaction data of rhesus macaques were obtained from the STRING database (version 10.5). Based on the expression data and protein interactions in the rhesus macaque brain, we established the gene expression networks and analysed their gene expression characteristics and topological properties.

**Functional connectivity of adult rhesus macaque brains**. The monkey fMRI data were from PRIME-DE, an open resource for non-human primate imaging (http://fcon_1000.projects.nitrc.org/indi/indiPRIME.html)[70]. The neuroimaging data were collected from a group of 12 male anaesthetized rhesus macaque monkeys at the University of Western Ontario[71,72]. The resting-state experiments were conducted on a 7 T MRI scanner equipped with a 40-cm gradient coil set of 80 mT/m strength, and a custom-made 24-channel phased array receive coil with an 8-channel transmit coil was used. Resting-state images were acquired using a 2-dimensional multiband and EPI sequence. The preprocessing steps were performed using SPM8 for each monkey. The first 10 time points were dismissed to account for magnetic saturation. Then, the following steps were performed: (1) slice timing correction; (2) motion correction; (3) normalization with the INIA19 template (1.0-mm isotropic voxels)[73]; (4) spatial smoothing using a 2-mm full width half-maximum Gaussian kernel; (5) linear detrending and bandpass temporal filtering (0.01–0.3 Hz); (6) regression of nuisance variables, including the six parameters obtained by rigid body head motion correction, global signal, and their first temporal derivatives. Based on the D99 template of the macaque brain[37], we selected significant functional correlations to obtain the binary $304 \times 304$ symmetric connectivity matrix of the functional connectivity network in macaque brains and standardized it with a Z-transform (Supplementary Data 9).

**Statistics and reproducibility**. The gene expression heterogeneity indices in different brain regions were expressed as mean ± standard deviation and compared using Student's $t$-tests. The correlation between different indices was analyzed using the Pearson method in the Matlab software. All tests were two-sided and $p < 0.05$, was considered to indicate statistical significance.

**Reporting summary**. Further information on research design is available in the Nature Research Reporting Summary linked to this article.

## Data availability
The expression data used in this study are available via the Allen Institute for Brain Atlas (see http://brain-map.org/). The interactions in human, rhesus macaque and mouse are available in Supplementary Data 8, STRING and BIOGRID database. Resting-state fMRI data of humans, rhesus macaques, and mice can be found in http://www.humanconnectome.org/documentation/S500/, http://fcon_1000.projects.nitrc.org/indi/indiPRIME.html, and central.xnat.org, respectively. The authors declare that the data supporting the findings of this study are available within the article, its supplementary information, and upon request.

## Code availability
Custom MATLAB code to analyse the gene expression characteristics and the topological properties of networks in this work is available at https://github.com/angelnudt/gene-expression-heterogeneity-analysis.git.

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

## Acknowledgements
We would like to thank Prof. Lin Xu in the laboratory of Learning and Memory Kunming Institute of Zoology, Chinese Academy of Sciences for his excellent advice. This study was supported by the National Science Foundation of China (61722313, 62036013, and 61773391) and the Fok Ying Tung Education Foundation (161057), and the Science & Technology Innovation Program of Hunan Province (2018RS3080).

## Author contributions
W.L., L.-L.Z., and D.H. designed the experiments. W.L., L.-L.Z., and H.S. performed the primary analyses, and supporting analyses were performed by Z.-T.Z., and D.H. The graphics and network analyses were performed by W.L., and Z.-T.Z. L.-L.Z. performed the functional MRI analysis. W.L., L.-L.Z., H.S., and D.H. wrote the initial draft, and all authors contributed to, read, and approved the final manuscript.

## Competing interests
The authors declare no competing interests.
