## [Peer Review File · Communications Biology]

Reviewers' comments:

Reviewer #1 (Remarks to the Author):

In the current work, the authors evaluated the association between gene expression heterogeneity and functional organisation of the human cortex. To do so they compared genetic expression in humans with functional connectivity and expression and functional connectivity in macaques and mice. They find that gene expression heterogeneity may be heightened in humans, in particular in association regions.

Overall I think the work has some potential, though it is hard to evaluate as methods are not very easy to follow and lack detail of key steps. Observed differences may be accounted for by sampling rates in the AHBH as a function of cortical hierarchy. Moreover, it may be helpful to further embed this work in the literature, to help understand what aspects are confirmatory of earlier findings and what aspects are novel. For example, this paper may be interesting:

<https://www.biorxiv.org/content/10.1101/2020.06.21.163915v2.abstract> I hope my comments are of help.

1. Possibly have another look at the writing and text of the manuscript, there are various passages that are difficult to understand due to wording choices (origin age of evolutionary history for example, but also subcortical area in the abstract should be areas, and there is also a typo in Figure 1).

2. Figure 1 and the distribution of percentages is rather difficult to interpret, particularly as the colours do not really vary linearly. Possibly it would help with interpretation if this would change (also the case for other figures).

3. What does it mean when there is higher variability but also reduced genetic expression and higher specificity in neocortex relative to archi- and subcortex?

4. How does measured expression related to number of measurements/samples? Why was not the consistent average across samples in the left-hemispheres computed for gene expression? How consistent was this pattern across donors?

5. I believe that the comparison with mice and macaques is very interesting but at the moment lacks depth and direct comparison to interpret what it means. For example, it seems that human findings are replicated in macaques?

6. The discussion is a bit thin to fully grasp the meaning of the current results and it would be helpful to embed them better in the literature, in particular as a lot of related work has been done with the Allen Human Brain Atlas.

7. Methods on the preprocessing of gene expression of donors and how to was sampled to what atlas are lacking

8. Why where functional connectivity matrices not Fishers Z transformed?

9. It is unclear why code is only available upon request.

Reviewer #2 (Remarks to the Author):

The present manuscript addresses an interesting and timely question, namely how the functional topography of brain networks and the cross-region variation in gene expression patterns are related. The authors use a variety of existing sources to establish gene expression across the brain and to determine functional networks in the brain. They focus first on the human brain, but also perform some analyses on the mouse brain.

I think this work is potentially interesting, but I'm afraid to say the presentation leaves much to be desired. I have four main criticisms:

First, the language is awkward and often grammatically incorrect. I realise science consists mostly of non-native English speakers, but when it impedes interpretation of the work, it should be addressed. I found it very hard to judge exactly what was being done and why.

Second, the rationale of the approach is often not made clear. Yes, the authors want to understand

the relationship between functional topography and gene expression, but the usefulness of their methods is often not communicated. For instance, they calculate a number of network measure on the gene expression networks. But I fail to see what these measures on gene expression data tell us about the brain. I understand what path length means in functional data, but in gene expression data it is much more arbitrary. Similarly, a number of quantifications of entropy are made, without it being clear what this tell us about brain organization.

Third, as distinct from why certain methods are used, often the details of the methods are not clear. For instance, in the methods section they suddenly talk about dN/dS values, without explaining the terms.

Finally, the work needs to be better embedded within the neuroscientific literature. There are lot of instances of reserve inference ("This brain regions does this and therefore it's correct that we find this") which is a bit disturbing. Also, terms like "cognition" are used in a rather vague and unspecified way.

I'm sorry to be negative about this manuscript. I applaud the general idea, but I fear the present manuscript does not teach me much news about how the brain is organized. It therefore fails in its communicative goal.

Response to Comments of Reviewer #1

In the current work, the authors evaluated the association between gene expression heterogeneity and functional organisation of the human cortex. To do so they compared genetic expression in humans with functional connectivity and expression and functional connectivity in macaques and mice. They find that gene expression heterogeneity may be heightened in humans, in particular in association regions.

Overall I think the work has some potential, though it is hard to evaluate as methods are not very easy to follow and lack detail of key steps. Observed differences may be accounted for by sampling rates in the AHBH as a function of cortical hierarchy. Moreover, it may be helpful to further embed this work in the literature, to help understand what aspects are confirmatory of earlier findings and what aspects are novel. For example, this paper may be interesting: <https://www.biorxiv.org/content/10.1101/2020.06.21.163915v2.abstract>. I hope my comments are of help.

1. Possibly have another look at the writing and text of the manuscript, there are various passages that are difficult to understand due to wording choices (origin age of evolutionary history for example, but also subcortical area in the abstract should be areas, and there is also a typo in Figure 1).

Response: Thank you for your comment. We have corrected these sentences and hired **American Journal Experts (AJE)** (<http://www.journalexerts.com>) to thoroughly edit the manuscript.

2. Figure 1 and the distribution of percentages is rather difficult to interpret, particularly as the colours do not really vary linearly. Possibly it would help with interpretation if this would change (also the case for other figures).

Response: Thank you very much for your suggestion. In preparing this manuscript, we have tried several ways to show the mapping results of measures in brain areas, including the way that the colours vary linearly. However, we found the measures of different cortex have different distribution range, and there was a certain intersection between the measure values of different cortex. For example, the variation range of HKG percentage in the subcortex is much greater than that of the neocortex and archicortex, as shown in Figure 1. Finally we selected the cutoffs of these measures which can be most distinguished among the neocortex, archicortex and subcortex, in order to display the change trend of these measure in different brain areas.

Some descriptions were added in the legend of Figure 1 and Figure 2 as follow:

‘The HKG percentage cut-off to best distinguish the neocortex and the archicortex and subcortex of Brain #1 was set at 32.45% and 35.90%, respectively.’

‘The mean path length cut-offs to best distinguish the neocortex, archicortex and subcortex of Brain #1 are 2.8073 and 2.7939, respectively, while the cut-offs of the clustering coefficient are 0.2065 and 0.2089, respectively.’

3. What does it mean when there is higher variability but also reduced genetic expression and higher specificity in neocortex relative to archi- and subcortex?

Response: Thank you for your comment. The current results indicate that the genes expressed in the neocortex tend to have higher variability, lower expression levels and higher expression specificity than those in the archicortex and subcortex. This is because there are higher proportion

of non-HKGs, including specific genes, expressed in the neocortex than in archicortex and subcortex. As reported, HKGs are involved in basic cell maintenance and, therefore, are expected to maintain constant expression levels in all cells and conditions and have a relatively low evolutionary rate. On the contrary, specific genes have more variable expression patterns and lower average expression levels, compared with housekeeping genes. We found a similar trend of gene expression in cerebral cortex of human brain, with higher expression variability of genes in multimodal association cortex than in unimodal primary cortex.

In the revised manuscript, we have added the relevant descriptions to the section of Discussion (Page 16):

“Our results indicated that the genes expressed in the neocortex tend to have higher variability, lower gene expression levels and higher expression specificity than those in the archicortex and subcortex. This is because there is a higher proportion of non-HKGs, including specific genes, expressed in the neocortex than in the archicortex and subcortex. As reported, HKGs are involved in basic cell maintenance and, therefore, are expected to maintain constant expression levels in all cells and conditions⁴² and have a relatively low evolutionary rate⁴³. In contrast, specific genes have more variable and reduced expression patterns. We found a similar trend of gene expression in the cerebral cortex of the human brain, with higher expression variability of genes in the multimodal association cortex than in the unimodal primary cortex.”

4. How does measured expression related to number of measurements/samples? Why was not the consistent average across samples in the left-hemispheres computed for gene expression? How consistent was this pattern across donors?

Response: Thank you for your insightful comment. The gene expression data were obtained from six adult brains, for a total of 3,702 brain samples, and the number of measured samples in different donors are different. We usually take Brain #1 as the example to show the analysis results, since there are 946 samples measured in Brain #1, which is the highest in all donors. Within the same donor, the HKG percentage and gene expression number of samples are comparable. We take the gene expression number of a sample as the number of genes significantly expressed than the microarray background in this sample, and the HKG percentage of a sample as the percentage of the genes expressed in almost all samples occupying the expression gene number of this sample. Considering these two indicators may change cross donors, we only compare them within donors. Although the values of these indicators are different in different donors, their distribution from neocortex to archicortex and subcortex are basically the same in all donors. In the revised manuscript, we have added the relevant descriptions (Page 5):

“Considering individual differences and different numbers of samples among donors, we analysed and compared the gene expression networks between human brain samples. Based on the gene expression networks, we extracted the percentages of HKGs and specific genes that were expressed in almost all brain samples²³ and expressed in only one or two samples²⁴, respectively. Although the values of these indicators are variable in different donors, their distribution in the neocortex to archicortex and subcortex are basically the same across all donors. We took Brain #1 as an example to show the analysis results, since it has the highest number of samples of all donors.”

In the later part of investigating the heterogeneity of internal structures in the neocortex, we

integrated the gene expression data of brain samples from six donors. The results from the integrated brain data were in accord in the analysis results from single donor, that is greater heterogeneity of gene expression in neocortex than those in archicortex and subcortex.

5. I believe that the comparison with mice and macaques is very interesting but at the moment lacks depth and direct comparison to interpret what it means. For example, it seems that human findings are replicated in macaques?

Response: Thank you for your good suggestion. We make some evolutionary analyses across species to investigate whether the gene expression characteristics of rhesus macaque and mouse brains is similar to that of the human brain. As we expected, the results about the gene expression heterogeneity of rhesus macaque and mouse brains are consistently with that of human brain. Furthermore, we found some differences of gene expression in the cerebellum between humans and mice. This finding is based on the fine sampling of mouse brain and the measurement of gene expression. Unfortunately, we did not find a gene expression dataset of the macaque brain with enough fine sampling, especially in the cerebellum, to make a detailed comparison of the gene expression characteristics of these three species. Interspecies regional-matched differences will be analyzed when more datasets are available.

In the revised manuscript, we have added the relevant descriptions (Page 18):

“Due to the limitation of the sampling fineness of the gene expression dataset in the macaque brain, we have not yet found unique expression characteristics in the macaque brain compared with those in the human brain and mouse brain.”

6. The discussion is a bit thin to fully grasp the meaning of the current results and it would be helpful to embed them better in the literature, in particular as a lot of related work has been done with the Allen Human Brain Atlas.

Response: Thank you so much for your good suggestion. In the revised manuscript, we compared our results with those in the literature and added more detailed discussion in the Discussion section. Especially, we enhanced the comparison of our work with related work based on the Allen Human Brain Atlas.

In the Discussion section of the revised manuscript, we have (Page 16-18):

“In particular, the increased heterogeneity of gene expression in multimodal association areas may potentially regulate the specialization of higher-order cognitive functions in human brain evolution, compatible with prior observations of high expression of evolution-related genes in these brain areas⁷ and increased transcriptional complexity in the frontal lobe of the human brain⁴¹.”

“Our results indicated that the genes expressed in the neocortex tend to have higher variability, lower gene expression levels and higher expression specificity than those in the archicortex and subcortex. This is because there is a higher proportion of non-HKGs, including specific genes, expressed in the neocortex than in the archicortex and subcortex. As reported, HKGs are involved in basic cell maintenance and, therefore, are expected to maintain constant expression levels in all cells and conditions⁴² and have a relatively low evolutionary rate⁴³. In contrast, specific genes have more variable and reduced expression patterns. We found a similar trend of gene expression in the cerebral cortex of the human brain, with higher expression variability of genes in the multimodal association cortex than in the unimodal primary cortex.”

“Evolutionary changes in structures of the human brain relative to other mammalian brains can

arise from the emergence of new genes but more from quantitative expression changes in mRNA⁴⁴. Comparative transcriptome studies of the human and chimpanzee brain indicated that the acceleration signal is clearly more pronounced in the PFC, a region involved in high-order cognitive processes, than in other brain regions⁴⁵. ”

41. Konopka, G., Friedrich, T., Davis-Turak, J. et al. Human-specific transcriptional networks in the brain. *Neuron* **75**(4), 601-617 (2012).

42. Eisenberg, E., Levanon, E.Y. Human housekeeping genes, revisited. *Trends Genet.* **29**(10), 569-574 (2013).

43. Subramanian, S., Kumar, S. Gene expression intensity shapes evolutionary rates of the proteins encoded by the vertebrate genome. *Genetics* **168**, 373-381 (2004).

44. Wilson, M. D., Odom, D. T. Evolution of transcriptional control in mammals. *Curr Opin Genet Dev* **19**, 579-585 (2009).

45. Somel, M., Rohlf, R., Liu, X. Transcriptomic insights into human brain evolution: acceleration, neutrality, heterochrony. *Current Opinion in Genetics & Development* **29**, 110-119 (2014).

7. *Methods on the preprocessing of gene expression of donors and how to was sampled to what atlas are lacking.*

Response: Thank you for your kind reminder. In the revised manuscript, we have added some description in the section of Methods:

“Publicly available gene expression data from six human postmortem donors (one female and 5 males), aged 24-57 years (M = 42.5, SD = 13.38), were obtained from the AHBA and downloaded after the updated microarray normalization pipeline implemented in March 2013 (<http://human.brain-map.org>). The dataset consists of normalized expression data detected by 58,692 probes taken from 3,702 spatially distinct tissue samples. Before tissue dissection, donor brains underwent anatomical MRI scanning and alignment to MNI space by the Allen Institute. Available samples were prepared for microarray expression analyses by the Allen Institute via macrodissection for cortical areas or laser dissection for subcortical regions²². For additional information on structural imaging data as well as microarray preprocessing and normalization procedures, refer to the AHBA technical white paper (help.brain-map.org/display/humanbrain/Documentation). The expression level of each gene from all probes was averaged if there were multiple probes for the same gene. The resulting dataset contained 29,180 unique mRNA probes, providing transcriptional data of human brains.”

8. *Why where functional connectivity matrices not Fishers Z transformed?*

Response: Thank you for your kind reminder. Actually, the functional connectivity matrices were Fisher’s Z-transformed in our study. We are sorry for our carelessness to forget to describe this step in the Methods section. In the revised manuscript, we have added the relevant descriptions (Page 20):

“We evaluated the functional connectivity between each pair of regional averaged time courses using Pearson’s correlation coefficient and then standardized the functional connectivity matrix with Fisher’s Z-transform.”

9. *It is unclear why code is only available upon request.*

Response: Thank you for your comment. We have provided the code for analyzing the gene

expression characteristics and the topological properties of networks in this study at the website <https://github.com/angelnudt/gene-expression-heterogeneity-analysis.git>.

We appreciate your constructive comments very much and hope that the corrections will be met with your approval.

Response to Comments of Reviewer #2

The present manuscript addresses an interesting and timely question, namely how the functional topography of brain networks and the cross-region variation in gene expression patterns are related. The authors use a variety of existing sources to establish gene expression across the brain and to determine functional networks in the brain. They focus first on the human brain, but also perform some analyses on the mouse brain.

I think this work is potentially interesting, but I'm afraid to say the presentation leaves much to be desired. I have four main criticisms:

1. First, the language is awkward and often grammatically incorrect. I realise science consists mostly of non-native English speakers, but when it impedes interpretation of the work, it should be addressed. I found it very hard to judge exactly what was being done and why.

Response: Thank you for your comment. We have corrected these sentences and hired **American Journal Experts (AJE)** (<http://www.journalexerts.com>) to thoroughly edit the manuscript.

2. Second, the rationale of the approach is often not made clear. Yes, the authors want to understand the relationship between functional topography and gene expression, but the usefulness of their methods is often not communicated. For instance, they calculate a number of network measure on the gene expression networks. But I fail to see what these measures on gene expression data tell us about the brain. I understand what path length means in functional data, but in gene expression data it is much more arbitrary. Similarly, a number of quantifications of entropy are made, without it being clear what this tell us about brain organization.

Response: Thank you for your comment. Topology indices of gene expression networks in brain structures offer possibilities to understand how genes work together to perform diverse functions. We hypothesized that the heterogeneity of gene expression in different brain regions will affect the topological properties of gene expression networks. Thus, we computed several topological measures of gene expression networks and expected to put the changes of gene expression heterogeneity observed into a systems level context. Our results showed that the expression proportions of non-HKGs in the different brain structures are significantly correlated with the network measure on the gene expression networks. The gene expression networks in the neocortex have longer mean path lengths, smaller clustering coefficients and larger eigen entropies, than those in the archicortex and subcortex. This indicated the gene expression networks in the neocortex are sparser, contain less clusters and are more disorderly, compared with those in the archicortex and subcortex. Such an organizational structure possibly contributes to the genes expressed in neocortex performing more diverse functions than those in the archicortex and subcortex at the expense of partial efficiency.

We have provided more interpretations for the network measures in the revised manuscript in the section of Discussion and Methods:

“The gene expression networks in the neocortex are sparser, contain fewer clusters and are more disorderly than those in the archicortex and subcortex. Such an organizational structure possibly contributes to the genes expressed in neocortex performing more diverse functions than those in the archicortex and subcortex at the expense of partial efficiency.”

“Distance in networks is measured with the path length, and the shortest path, the path with the

smallest number of links between the selected nodes, has a special role. The mean path length represents the average of the shortest paths between all pairs of nodes and offers a measure of a network's overall navigability⁵⁵. Cellular functions are likely to be carried out in a highly modular manner. The average clustering coefficient characterizes the overall tendency of the nodes to form clusters or groups⁵⁶. The closer the local clustering coefficient is to 1, the more likely that the network will form clusters. The eigen entropy of networks was defined as the entropy of the normalized largest eigen vector of an adjacency matrix²⁷. For a network with n genes, its adjacency matrix can be represented as $A_{ij}(i, j = 1 \sim n)$, which describes the interacting relationships of the genes in the networks. If gene i interacts with gene j , $A_{ij} = 1$; otherwise, $A_{ij} = 0$. We assumed that the eigenvector of matrix A corresponding to the largest eigenvalue λ_k

is r_k . By dividing by the sum of all eigen vectors, we normalized this vector to obtain its energy

concentration, i.e., $I_k = \frac{r_k}{\sum_{i=1}^n r_k}$. The eigen entropy of this network is defined as

$S_p = -\sum_{k=1}^n I_k \ln I_k$, which can reflect the degree of ordering in each network. A smaller eigen

entropy of a network corresponds to a higher degree of ordering of the network.”

3. Third, as distinct from why certain methods are used, often the details of the methods are not clear. For instance, in the methods section they suddenly talk about dN/dS values, without explaining the terms.

Response: Thank you for your kind reminder. In the revised manuscript, we not only provided the detailed preprocessing procedures of gene expression data of human brains, but also more descriptions about the gene characteristics analysis, topological property calculation and functional connectivity of human, rhesus macaque and mouse brains in section of Methods.

In the revised manuscript, we have added the relevant descriptions (Page 20):

“The selective pressure is assumed to be defined by the ratio dN/dS . dS represents the synonymous substitution rate (changing the amino acid), and dN represents the nonsynonymous substitution rate (keeping the amino acid). Under purifying selection, natural selection prevents the replacement of amino acids, so dN will be lower than dS . Values of $dN/dS < 1$, $= 1$, and > 1 indicate negative purifying selection, neutral evolution, and positive selection, respectively.”

4. Finally, the work needs to be better embedded within the neuroscientific literature. There are lot of instances of reserve inference (“This brain regions does this and therefore it's correct that we find this”) which is a bit disturbing. Also, terms like “cognition” are used in a rather vague and unspecified way.

I'm sorry to be negative about this manuscript. I applaud the general idea, but I fear the present

manuscript does not teach me much news about how the brain is organized. It therefore fails in its communicative goal.

Response: Thank you for your comment. We have carefully revised the manuscript to avoid giving too strong conclusion and disturbing inference. Previous studies have suggested some possible links between molecular function and large-scale network organization of the human connectome. Our work further found functional orderly topography of brain networks associated with some measures of gene expression heterogeneity. Our results may provide insights into the molecular bases of brain organization, but how the basic layout of cortical areas was formed and the underlying molecular mechanism need to further investigation.

In the revised manuscript, we compared our results with those in the literature and added more detailed discussion in the Discussion section (Page 15~18).

“In particular, the increased heterogeneity of gene expression in multimodal association areas may potentially regulate the specialization of higher-order cognitive functions in human brain evolution, compatible with prior observations of high expression of evolution-related genes in these brain areas⁷ and increased transcriptional complexity in the frontal lobe of the human brain⁴¹.”

“Our results indicated that the genes expressed in the neocortex tend to have higher variability, lower gene expression levels and higher expression specificity than those in the archicortex and subcortex. This is because there is a higher proportion of non-HKGs, including specific genes, expressed in the neocortex than in the archicortex and subcortex. As reported, HKGs are involved in basic cell maintenance and, therefore, are expected to maintain constant expression levels in all cells and conditions⁴² and have a relatively low evolutionary rate⁴³. In contrast, specific genes have more variable and reduced expression patterns. We found a similar trend of gene expression in the cerebral cortex of the human brain, with higher expression variability of genes in the multimodal association cortex than in the unimodal primary cortex.”

“Evolutionary changes in structures of the human brain relative to other mammalian brains can arise from the emergence of new genes but more from quantitative expression changes in mRNA⁴⁴. Comparative transcriptome studies of the human and chimpanzee brain indicated that the acceleration signal is clearly more pronounced in the PFC, a region involved in high-order cognitive processes, than in other brain regions⁴⁵.”

41. Konopka, G., Friedrich, T., Davis-Turak, J. et al. Human-specific transcriptional networks in the brain. *Neuron* **75**(4), 601-617 (2012).

42. Eisenberg, E., Levanon, E.Y. Human housekeeping genes, revisited. *Trends Genet.* **29**(10), 569-574 (2013).

43. Subramanian, S., Kumar, S. Gene expression intensity shapes evolutionary rates of the proteins encoded by the vertebrate genome. *Genetics* **168**, 373-381 (2004).

44. Wilson, M. D., Odom, D. T. Evolution of transcriptional control in mammals. *Curr Opin Genet Dev* **19**, 579-585 (2009).

45. Somel, M., Rohlf, R., Liu, X. Transcriptomic insights into human brain evolution: acceleration, neutrality, heterochrony. *Current Opinion in Genetics & Development* **29**, 110-119 (2014).

We appreciate your constructive comments very much and hope that the corrections will be met with your approval.

Reviewers' comments:

Reviewer #1 (Remarks to the Author):

I thank the authors for their careful revision. However, based on the letter and the manuscript, it seems the authors did not consider <https://abagen.readthedocs.io/en/stable/> as a tool to investigate the transcriptomic expression in humans. Was there a particular reason not to do this? Based on my understanding of transcriptomic data from ABHA, there are a lot of potential preprocessing steps that may influence final results (see also: Arnatkevičiūtė et al., 2019), thus I would recommend using or at least validating current results using this established toolbox and workflow.

Reviewer #2 (Remarks to the Author):

I thank the authors for their revision. They helped me clarify the scope of the paper, the methods, and the overall claims. I think the approach is interesting and some of the results, although admittedly we are early in this field's development, are intriguing. However, I do think the manuscript can benefit from some additional changes.

I must admit I sometimes find the terminology regarding both evolution and anatomy in the manuscript a bit naive. For instance, some of the evolutionary terminology is quite strange. This starts with the abstract where the authors state that "The human cerebral cortex has evolved much more rapidly relative to nonhuman primates and rodents, leading to a functional orderly topography of brain networks." I think it's formally incorrect to say the human brain evolved more rapidly, rather there is more rapid evolutionary change in the human brain. This is a subtle, but important distinction. Also, I would argue that it is not necessarily the vast rate of evolutionary change that is the cause of the functional orderly topography.

In the introduction, the authors state "According to the origin age of evolutionary history". I'm afraid I cannot parse that sentence.

I would also caution the use of the term 'evolutionary sequence' when describing subcortex, archicortex and neocortex. Yes, neocortex is a more recent invention, but it is homologous to the dorsal cortex present in the mammalian-reptilian ancestor, and the gene expression architecture of the dorsal cortex is, I believe, still unknown.

Similarly, the presentation of hypotheses seems quite simplistic: "Compared to the archicortex and subcortex, the neocortex in the human brain performs diverse and complex functions and thus likely requires higher levels of gene expression in addition to HKGs." I think we can be more specific about what is different about the neocortex than saying "it performs complex functions".

Similar comments can be made about statements about cognition and cognitive functions. I still feel that notion that 'cognition' is the job of the neocortex and the rest is the job of the allo- and subcortex is rather outdated. Indeed, cognitive functions are due to specific cortical-subcortical and cortical-cerebellar loops. I think the manuscript should be updated to reflect this. Also, sentences like "PFC, a region involved in higher-order cognitive processes" seem from an undergraduate essay.

Sometimes this naive terminology hides the quite interesting findings of the manuscript. A case in point of this is the interesting finding of strong differences between the human cerebellar nuclei and the cerebellar lobules and vermis—the latter of which are strongly connected to the neocortex and also show smaller percentage HKG. I wonder if this could also explain the differences found between the human and mouse cerebellum. Was there less sampling of lobules/vermis in the mouse because there is less neocortex in the mouse, i.e., do the results actually follow the pattern

that the author highlight (higher cognitive regions have less HKG) both within and across species?

By the way, do the authors mean "the mean path length and eigen entropy of the gene expression networks in the isocortex (4.09 ± 0.09 and 5.13 ± 0.10 , respectively) were significantly higher than those in the cerebral nuclei (3.83 ± 0.10 and 4.96 ± 0.05 , respectively)" or do they mean to say "cerebellar nuclei" here?

I wonder if the authors have looked into the sampling of different cortical layers of the gene expression? Presumably, the more layers in the neocortex could be partially responsible for the greater heterogeneity of the gene expression profiles? This would help explain, but not invalidate, the results obtained by the authors.

I'm very sorry, but I still do not understand the network measures use on the gene expression data, because I do not understand what example the matrices consist of. I'm talking about the section of the methods "Topological properties of networks". Is the data a matrix of size $\text{brain_regions_in_network} * \text{genes}$ or what? What does I mean when the authors say if gene i "interacts" with gene j?

Although the manuscript has been edited by a commercial English language agent, some errors remain. For instance:

At the end of the introduction, the sentence stating the goal of the study is not grammatically correct: "Here, we speculated the heterogeneity of gene expression in brain structures, including the proportion of housekeeping genes occupying all expressed genes and topology indices of gene expression networks. "

"The multimodal association cortex, including frontoparietal control, attention, and default networks, showed reduced gene expression similarity within networks than the visual, motor, and limbic networks" -> "than" should be replaced with "compared to".

Response to Comments of Reviewer #1

I thank the authors for their careful revision. However, based on the letter and the manuscript, it seems the authors did not consider <https://abagen.readthedocs.io/en/stable/> as a tool to investigate the transcriptomic expression in humans. Was there a particular reason not to do this? Based on my understanding of transcriptomic data from ABHA, there are a lot of potential preprocessing steps that may influence final results (see also: Arnatkevičiūtė et al., 2019), thus I would recommend using or at least validating current results using this established toolbox and workflow.

Response:

Thank you very much for your comment. Considering that choice of processing steps and parameters can have a potential influence on the statistical outcomes of research with the AHBA, we used abagen as a powerful tool to investigate the influence of methodological variability on our results. We varied the methods of gene normalization, sample normalization and probe selection and took other parameters as default values and computed the heterogeneity index of brain regions based on the Desikan-Killiany atlas. The results are consistent under different standardizing workflows. In 10 distinct processing pipelines, the heterogeneity of gene expression in the neocortex was found to be significantly greater than those in the archicortex and subcortex, with the regions in neocortex tend to have lower standard errors of gene expression levels than those in archicortex and subcortex.

In the revised manuscript, we have added some descriptions in the section of Discussion (Page 16):

“Based on the abagen toolbox⁴¹, we examined the influence of methodological variability on our results and found that the change trend of the expression heterogeneity from the neocortex to the subcortex is consistent under different standardizing workflows (see Supplementary Method and Supplementary Table 5).”

41. Markello, RD, Arnatkevičiūtė, A, Poline, J-B, Fulcher, BD, Fornito, A, & Misic, B. Standardizing workflows in imaging transcriptomics with the abagen toolbox. *Elife* 10, e72129 (2021).

We appreciate your constructive comments very much and hope that the corrections will be meet with your approval.

Response to Comments of Reviewer #2

1) I thank the authors for their revision. They helped me clarify the scope of the paper, the methods, and the overall claims. I think the approach is interesting and some of the results, although admittedly we are early in this field's development, are intriguing. However, I do think the manuscript can benefit from some additional changes.

I must admit I sometimes find the terminology regarding both evolution and anatomy in the manuscript a bit naive. For instance, some of the evolutionary terminology is quite strange. This starts with the abstract where the authors state that "The human cerebral cortex has evolved much more rapidly relative to nonhuman primates and rodents, leading to a functional orderly topography of brain networks." I think it's formally incorrect to say the human brain evolved more rapidly, rather there is more rapid evolutionary change in the human brain. This is a subtle, but important distinction. Also, I would argue that it is not necessarily the vast rate of evolutionary change that is the cause of the functional orderly topography.

In the introduction, the authors state "According to the origin age of evolutionary history". I'm afraid I cannot parse that sentence.

Response:

Thank you very much for your suggestion. We have corrected these sentences and carefully revised the manuscript by using the terminology regarding evolution in the literature.

In the revised manuscript, the sentence "The human cerebral cortex has evolved much more rapidly relative to nonhuman primates and rodents, leading to a functional orderly topography of brain networks." was revised as "The human cerebral cortex is vastly expanded relative to nonhuman primates and rodents, leading to a functional orderly topography of brain networks." "According to the origin age of evolutionary history" was revised as "As the fruit of billions of years of evolution".

2) I would also caution the use of the term 'evolutionary sequence' when describing subcortex, archicortex and neocortex. Yes, neocortex is a more recent invention, but it is homologous to the dorsal cortex present in the mammalian-reptilian ancestor; and the gene expression architecture of the dorsal cortex is, I believe, still unknown.

Response:

Thank you for your comment. We have carefully revised the manuscript to avoid giving such vague or inaccurate statements.

In the revised manuscript, we have revised the sentence "Our results revealed that the orderness of the gene expression networks in the brain regions presented a downwards trend" to "Our results revealed that the orderness of the gene expression networks in the brain regions presented a downwards trend from the subcortex to the archicortex to the neocortex, consistent in all six brains."

3) Similarly, the presentation of hypotheses seems quite simplistic: "Compared to the archicortex and subcortex, the neocortex in the human brain performs diverse and complex functions and thus likely requires higher levels of gene expression in addition to HKGs." I think we can be more specific about what is different about the neocortex than saying "it performs complex functions".

Response:

Thank you for your comment. Although the neocortex interacts extensively with other parts of the brain, none of the human abilities are possible without neocortex.

This sentence has been revised as “The neocortex is the seat of higher cognitive functions, and involved in complex cognitive functions through cortical circuits, cortical-subcortical and cortical-cerebellar circuits and thus likely requires the expression of more genes other than HKGs.”

4) Similar comments can be made about statements about cognition and cognitive functions. I still feel that notion that ‘cognition’ is the job of the neocortex and the rest is the job of the allo- and subcortex is rather outdated. Indeed, cognitive functions are due to specific cortical-subcortical and cortical-cerebellar loops. I think the manuscript should be updated to reflect this. Also, sentences like “PFC, a region involved in higher-order cognitive processes” seem from an undergraduate essay.

Response:

Thank you so much for your good suggestion. I quite agree with you that cognitive functions are due to specific cortical-subcortical and cortical-cerebellar loops. In this paper, we associated the gene expression with resting-state functional connectivity network of human brain to investigate the expression heterogeneity of densely connected regions. We found the brain regions densely connected to other regions in the functional connectivity networks are primarily located in the neocortex of the human brain and tend to exhibit relative great expression heterogeneity. This result is consistent to the notion that the neocortex possesses a mosaic of regions, central to its information-processing capabilities and involved in cognitive functions through cortical circuits, cortical-subcortical and cortical-cerebellar circuits.

In the revised manuscript, we have added some descriptions in the section of Result (Page 6) and Discussion (Page 18):

“The neocortex is the seat of higher cognitive functions, and involved in complex cognitive functions through cortical circuits, cortical-subcortical and cortical-cerebellar circuits and thus likely requires higher levels of gene expression in addition to HKGs.”

“The brain regions with greater expression heterogeneity densely connected to other regions in the functional connectivity networks, supporting that the neocortex possesses a mosaic of regions, central to its information-processing capabilities. ”

The sentence “PFC, a region involved in higher-order cognitive processes” has been revised as “ PFC, a region involved in high order, partly human-specific cognitive processes such as abstract thinking and planning”.

5) Sometimes this naive terminology hides the quite interesting findings of the manuscript. A case in point of this is the interesting finding of strong differences between the human cerebellar nuclei and the cerebellar lobules and vermis—the latter of which are strongly connected to the neocortex and also show smaller percentage HKG. I wonder if this could also explain the differences found between the human and mouse cerebellum. Was there less sampling of lobules/vermis in the mouse because there is less neocortex in the mouse, i.e., do the results actually follow the pattern that the author highlight (higher cognitive regions have less HKG) both within and across species?

Response:

Thank you for your comment. In this paper, the cerebellum of mouse was divided into 5 structures, including vermal regions, hemispheric regions, fastigial nucleus, interposed nucleus and dentate nucleus. Because the expression levels of genes in these structures were obtained by averaging across all voxels in the brain structures based on the expression intensity in each voxel, it can be ensured that the cerebellar lobules/vermis in the mouse will not be ignored due to the issue of sampling. In particular, vermal regions and hemispheric regions of mouse exhibited very high HKG proportion, with values of 50.09% and 49.78%. All the structures in mouse cerebellum exhibited high expression heterogeneity, different from those in human cerebellum. So far, the result suggests there are the differences between the expression heterogeneity of the human and mouse cerebellum. We will further study interspecies regional-matched differences in multiple species when more datasets are available.

6) *By the way, do the authors mean “the mean path length and eigen entropy of the gene expression networks in the isocortex (4.09 ± 0.09 and 5.13 ± 0.10 , respectively) were significantly higher than those in the cerebral nuclei (3.83 ± 0.10 and 4.96 ± 0.05 , respectively)” or do they mean to say “cerebellar nuclei” here?*

Response:

Thank you for your comment. As shown in the Allen Mouse Brain Atlas (<http://atlas.brain-map.org/atlas>), the cerebral nuclei, including striatum and pallidum, is a part of mouse cerebrum.

In the revised manuscript, we have added some descriptions of cerebral nuclei (Page 14):
“The cerebral nuclei, including striatum and pallidum, is a part of mouse cerebrum.”

7) *I wonder if the authors have looked into the sampling of different cortical layers of the gene expression? Presumably, the more layers in the neocortex could be partially responsible for the greater heterogeneity of the gene expression profiles? This would help explain, but not invalidate, the results obtained by the authors.*

Response:

Thank you for your comment. Our analysis was based on the dataset of AHBA, consisting of normalized expression data taken from 3,702 spatially distinct tissue samples from six human postmortem donors. The more layers in the neocortex than the archicortex and subcortex lead to the more samples extracted from the neocortex, which may be partially responsible for the greater heterogeneity of the gene expression profiles in the neocortex. To avoid the influence of data sampling bias, we analyzed the expression heterogeneity based on the multiresolution gene expression networks in human brain samples, and the structure and region levels in rhesus macaque and mouse brains. The results obtained from the different resolution networks and different individuals are consistent, implying that such gene expression patterns in the brain are robust and inherent.

8) *I’m very sorry, but I still do not understand the network measures use on the gene expression data, because I do not understand what example the matrices consist of. I’m talking about the section of the methods “Topological properties of networks”. Is the data a matrix of size `brain_regions_in_network*genes` or what? What does I mean when the authors say if gene i*

"interacts" with gene j?

Response:

Thank you for your comment. By integrating the expression data and large-scale protein interaction data, we established gene expression networks for brain samples. For each sample, we obtained a matrix corresponding to its gene expression network, where nodes represent the genes expressed in the sample and edges represent that their gene products can interact. Based on these matrices, we computed some typical topological indices of gene expression networks.

In the revised manuscript, we have added some relevant descriptions in the section of Methods (Page 20):

“For each sample, we obtained a matrix corresponding to its gene expression network, where nodes represent the genes expressed in the sample and edges represent that their gene products can interact.”

9) Although the manuscript has been edited by a commercial English language agent, some errors remain. For instance:

At the end of the introduction, the sentence stating the goal of the study is not grammatically correct: “Here, we speculated the heterogeneity of gene expression in brain structures, including the proportion of housekeeping genes occupying all expressed genes and topology indices of gene expression networks. ”

“The multimodal association cortex, including frontoparietal control, attention, and default networks, showed reduced gene expression similarity within networks than the visual, motor, and limbic networks” -> “than” should be replaced with “compared to”.

Response:

Thank you for your comment. We have corrected these sentences and thoroughly edit the manuscript.

The sentence “Here, we speculated the heterogeneity of gene expression in brain structures, including the proportion of housekeeping genes occupying all expressed genes and topology indices of gene expression networks. “ has been revised as follows:

“Here, we measured gene expression heterogeneity with two metrics, i.e., the proportion of housekeeping genes occupying all expressed genes and topology indices of gene expression networks.”

“The multimodal association cortex, including frontoparietal control, attention, and default networks, showed reduced gene expression similarity within networks than the visual, motor, and limbic networks” -> “than” has been replaced with “compared to”.

We appreciate your constructive comments very much and hope that the corrections will be meet with your approval.

Reviewers' comments:

Reviewer #1 (Remarks to the Author):

Dear authors,

thank you for your edits.

I would like to further comment on the reply to reviewer 2;

I believe still various points with respect to evolution and cognition are quite coarse or not sufficiently backed up by references.

For example:

"The neocortex is the seat of higher cognitive functions, and involved in complex cognitive functions through cortical circuits, cortical-subcortical and cortical-cerebellar circuits and thus likely requires the expression of more genes other than HKGs."

This is a quite strong statement, and it would help to provide references, and/or consider a more nuanced view.

Moreover, it may be that various of heterogeneity are due to methodological differences rather than biological effects? In particular, using graph theory to describe heterogeneity may be tricky, as brain size, individual variation in cortical regions and other factors may also influence these results.

Reviewer #2 (Remarks to the Author):

Thank you for addressing my remaining comments.

Response to Comments of Reviewer #1

I would like to further comment on the reply to reviewer 2;

I believe still various points with respect to evolution and cognition are quite coarse or not sufficiently backed up by references.

For example:

(1) "The neocortex is the seat of higher cognitive functions, and involved in complex cognitive functions through cortical circuits, cortical-subcortical and cortical-cerebellar circuits and thus likely requires the expression of more genes other than HKGs."

This is a quite strong statement, and it would help to provide references, and/or consider a more nuanced view.

Response:

Thank you very much for your kind suggestion. We have weakened the statement and included more references in the revised manuscript. In the Introduction Section, we have (Page 3-4):

“The highly consistent transcriptional architecture in neocortex is correlated with resting-state functional connectivity¹³, and genetic and evolutionary uncoupling of structure and function in different transmodal systems may support the emergence of complex forms of cognition¹⁵.”

In the Discussion Section, we have (Page 16):

“The neocortex is the seat of higher cognitive functions⁴⁶ and involved in complex cognitive functions through cortical circuits⁴⁷, cortical-subcortical⁴⁸ and cortical-cerebellar circuits⁴⁹. Previous studies reported that spatial patterns of gene expression may reflect the hierarchical organization⁵⁰ and spatial gradients of intrinsic dynamics in neocortex⁵¹.”

References:

12. Ardesch, D. J., et al. Evolutionary expansion of connectivity between multimodal association areas in the human brain compared with chimpanzees. *Proc. Natl Acad. Sci. USA* **116(14)**, 7101-7106 (2019).
15. Valk, S. L. et al. Genetic and phylogenetic uncoupling of structure and function in human transmodal cortex. *Nat Commun.* **13(1)**, 2341 (2022).
23. Joshi, C. J. et al. What are housekeeping genes? *PLoS Comput Biol.* **18(7)**, e1010295 (2022).
45. Florio, M., Huttner, W. B. Neural progenitors, neurogenesis and the evolution of the neocortex. *Development* **141(11)**, 2182-2194 (2014).
46. Harris, K. D., Shepherd, G. M. The neocortical circuit: themes and variations. *Nat Neurosci.* **18(2)**, 170-81 (2015).
47. Anderson, K. M., et al. Gene expression links functional networks across cortex and striatum. *Nat Commun.* **9(1)**, 1428 (2018).
48. Wagner, M. J., Luo, L. Neocortex-cerebellum circuits for cognitive processing. *Trends Neurosci.* **43(1)**, 42-54 (2020).
49. Hansen, J. Y. et al. Mapping gene transcription and neurocognition across human neocortex. *Nat Hum Behav.* **5(9)**, 1240-1250 (2021).
50. Shafiei, G. et al. Topographic gradients of intrinsic dynamics across neocortex. *Elife* **9**, e62116 (2020).
51. Su, A. I. et al. A gene atlas of the mouse and human protein-encoding transcriptomes. *Proc Natl Acad Sci U S A.* **101(16)**, 6062-7 (2004).
53. Herbet, G., Duffau, H. Revisiting the Functional Anatomy of the Human Brain: Toward a Meta-Networking Theory of Cerebral Functions. *Physiol Rev.* **100(3)**, 1181-1228 (2020).

(2) Moreover, it may be that various of heterogeneity are due to methodological differences rather than biological effects? In particular, using graph theory to describe heterogeneity may be tricky, as brain size, individual variation in cortical regions and other factors may also influence these results.

Response:

Thank you for your insightful comments. We hold that the influence of methodology differences on the gene expression heterogeneity is limited.

To investigate the influence of brain area size, we analyzed the heterogeneity based on the multiresolution gene expression networks in human, rhesus macaque and mouse brains. The results obtained from the different resolution networks and different species are consistent. The heterogeneity of gene expression in the neocortex was found to be significantly greater than those in the archicortex and subcortex, associated with the functional orderly topography of brain networks.

Previous findings have suggested the patterning for any given gene across structures in different individuals was often well conserved¹³. To eliminate the influence of individual differences among donors, we analyzed and compared the gene expression heterogeneity between brain samples in the same donor. In Brain #1, the HKG percentages in the neocortex were significantly lower than those in the archicortex and subcortex (both P-values<0.001 in two sample t-test). Similar results were obtained in the other five human brain samples.

Furthermore, we introduced topological indices of gene expression networks in brain structures to measure the heterogeneity of gene expression. Cellular network has topological robustness, which can be resilient against component failure, withstanding even the incapacitation of many of their individual components and many changes in external conditions⁶³. Previous studies applied weighted gene co-expression network analysis to build co-expression networks, so as to identify modules of co-regulated genes¹⁶ or examine the systems level organization of lineage-specific gene expression differences⁴⁵. Comparisons of human and chimpanzee brains on the basis of gene connectivity led to the observation that the overall conservation of gene coexpression modules between the species recapitulates evolutionary hierarchies, a relationship not evident from differential expression analysis⁶⁴. Our analyses showed that increased expression of non-HKGs in the neocortex was associated with the changes in the topological properties of the gene expression networks. The distribution of these topological indices in the neocortex to archicortex and subcortex was basically consistent across all donors (Supplementary Fig. 4).

Supplementary Fig. 4 The mean value and standard deviation of topological indices in the neocortex, archicortex and subcortex of Brain #1-6

In the Discussion section of the revised manuscript, we have (Page 18):

“Weighted gene co-expression network analysis was applied to build co-expression networks, so as to identify modules of co-regulated genes¹⁶ or examine the systems level organization of lineage-specific gene expression differences⁴⁵. Cellular network has topological robustness against accidental failures⁶³ and the analysis based on gene connectivity may observe the overall conservation of gene coexpression modules between the species⁶⁴. “The distribution of the topological indices of the gene expression networks in the neocortex to archicortex and subcortex was basically consistent across all donors (Supplementary Fig. 4).”

References:

13. Hawrylycz, M. et al. Canonical genetic signatures of the adult human brain. *Nat. Neurosci.* 18(12), 1832-1846 (2015).
16. Kang, H. J. et al. Spatio-temporal transcriptome of the human brain. *Nature* 478(7370), 483-489 (2011).
45. Konopka, G., Friedrich, T., Davis-Turak, J. et al. Human-specific transcriptional networks in the brain. *Neuron* 75(4), 601-617 (2012).
63. Barabási, A. L. & Oltvai, Z. N. Network biology: understanding the cell's functional organization. *Nature Reviews Genetics* 5, 101-113 (2004).
64. Oldham, M. C., Horvath, S., Geschwind, D. H. Conservation and evolution of gene coexpression networks in human and chimpanzee brains. *Proc Natl Acad Sci U S A.* **103(47)**, 17973-17978 (2006).

We appreciate your constructive comments very much and hope that the corrections will meet with your approval.